# Data Mining in Coal-Mine Gas Explosion Accidents Based on Evidence-Based Safety: A Case Study in China

**Jiaqi Hu, Rui Huang * and Fangting Xu**

School of Resources and Safety Engineering, Central South University, Changsha 410083, China
* Correspondence: huangrui@csu.edu.cn

**Abstract:** From an informatics perspective, decision-making failures in accident prevention are due to insufficient necessary safety evidence. Analyzing accident data can help in obtaining safety evidence. Currently, such a practice mostly relies on experts' judgement and experience, which are subjective and inefficient. Furthermore, due to the inadequate safety-related theoretical support, the sustainable safety of a system can hardly be achieved purposefully. To automatically explore and obtain latent safety evidence in coal-mine data, and improve the reliability and sustainability of coal-mine safety management, a novel framework of combining data mining technology and evidence-based safety (EBS) theory is proposed, and was applied to a coal gas explosion accident. First, the term frequency-inverse document (TF-IDF) and TextRank algorithms were fused to extract keywords, and keyword evolution word cloud maps from the time dimension were drawn to obtain keyword safety evidence. Then, on the basis of the latent Dirichlet allocation (LDA) model, the best safety evidence, such as accident causation topics and causation factors, were mined, and safety decisions were given. The results show that accident data mining, based on evidence-based safety, can effectively and purposefully mine the best safety evidence, and guide safety decision making to optimize safety management models and achieve sustainable safety.

**Keywords:** accident causation topics; best safety evidence; latent Dirichlet allocation (LDA); accident prevention; sustainable safety

## 1. Introduction

As a critical resource in the world, coal has a great impact on the global economy and the development of humanity [1,2]. It also remains the primary energy source in many countries [3]. However, the coal-mining industry is undoubtedly a typical high-risk industry, with a high frequency of injuries and deaths, and various incentives. Coal-mining work is considered to be one of the most dangerous operations in the world [4]. In underground coal-mining accidents, the most common causes are coal gas outbursts, methane explosions, coal-dust explosions, suffocation, and flooding [5,6]. Among all types of accidents in the process of coal mining, gas explosion accidents are the most serious, as they may cause numerous casualties and huge economic losses, and have severe negative social effects [7]. Over the past four decades, about three dozen explosions of air–methane mixtures, or hybrid mixtures of coal dust with methane and air, have occurred in Ukrainian coal mines, causing hundreds of people to be fatally injured [8]. Coal methane explosions have also been major mining accidents in Turkey, since 1982 [9]. As mining conditions become more complex, and mining depth and intensity continue to increase, the gas content and emissions from coal seams increase dramatically, and new changes in gas prevention and control emerge. As a result, there are challenges faced in the effective management and control of coal-mine gas explosion risks [10,11]. Therefore, it is of great significance to find new effective ways to help in coal-mine gas prevention and control.

To establish the circumstances of coal-mine gas explosion accidents and improve safety management in coal mines, the statistical analysis of accidents is necessary, and common.

On the basis of a large number of accident cases, it helps to explore the circumstances of accident occurrence through relevant principles and statistical methods, and this provides a scientific basis for accident prevention and control. Wang et al. [12] analyzed 29 cases of significant coal-mine accidents that occurred in China in 2016, with regard to accident types, occurrence time, occurrence location, and direct causes, and summarized six types of accidents, including gas explosions, and their statistical characteristics. Shahani et al. [13] built a statistical multimodel to analyze fatal underground coal-mine accidents in Pakistan from 2010 to 2018. Yin et al. [14] statistically analyzed the specifics affecting coal-mine gas explosions in China, in terms of human factors. Dursun et al. [15] statistically analyzed methane explosions in coal mines in Turkey during 2010–2017, and found that the death toll caused by methane explosions and other gas-related accidents was 68.34%. However, due to the complexity of coal-mine systems and the various causes of gas explosion accidents, the statistical analysis of accident data is way too macroscopic, and cannot fully reflect the accident characteristics and deeply grasp the circumstances of accident occurrence. Accident causation models are an important tool in systematic safety management. The earliest accident causation theory and model were proposed by Heinrich in 1931 [16], and more theories and models have been developed since then. Bhattacharjee et al. [17] analyzed a coal dust explosion disaster using the accident causation tree and Swiss cheese model to identify the root cause. Zhang et al. [18] proposed a universal human-error causation model, and applied it to gas explosion accidents in China. Fu et al. [19] proposed an action path and analytical steps of accidents based on the 24 model, and applied them to coal-mine gas explosion accidents. Sherin et al. [20] obtained 43 root causes of surface mine accidents through fault tree analysis based on the 3E model. These studies can, to some extent, reflect the causes of small-scale accidents, and provide accident prevention measures. However, most of the studies relied on experts' experience and judgment, which are very time-consuming and subjective, and cannot deal with large-scale accident cases in a short period of time. With the popularization of safety information technology, a large number of safety production datasets reflecting the essential laws of safety production, and the basic values of safety assurance, have been accumulated; it has become an important trend to dig out potential safety information from big data to assist in safety decision making [21]. In order to improve safety management more intelligently, an increasing number of new theories and automation techniques are being researched. Kobylianskyi et al. [22] introduced a smart-protection system that is triggered at the stage of hazard identification, improving decision-making adequacy to, thereby, improve the safety management of the system. Hughes et al. [23] introduced a semiautomated technique for classifying text-based close all reports from the UK railway industry. Comberti et al. [24] used an improved approach composed of SOM and numerical clustering to analyze occupational accident data, and obtained positive results in identifying critical accident dynamics. Tanguy et al. [25] used natural language processing and text mining techniques to mine information in the field of aviation to help in better classifying and retrieving information. Liao et al. [26] used the Apriori algorithm to mine the association rules of occupational injury factors in the construction industry in Taiwan. There are also studies [27–29] that applied deep-learning and machine-learning methods to analyze safety data in the construction, transportation, and chemical fields. However, due to a lack of safety-related theory, this may often lead to not being able to mine deeper information, and achieve the sustainable safety of the system.

Traditional statistical analysis often fails to reflect the accident characteristics of complex safety systems due to the macroscopic perspective of analysis. The processing of accident cases by building complex accident causation models often consumes a lot of time, and manual effort. Although new automated technologies can quickly process data, due to the lack of relevant safety theory support, the relevant research did not study the accident information from the perspective of the sustainable safety of the system, and the resulting information is not rich enough. Therefore, it is not able to dig deep into the relevant safety information, and it is not possible to refine the scientific evidence of accident causation; prevention measures often lead to failures in decision making due to the lack of scientific

evidence. It is, therefore, difficult to achieve sustainable safety. Overall, the research goal of this study is to find a pertinent and efficient method for automatically and deeply mining accident information.

The Evidence-Based Safety (EBS) theory was initially put forth by Wang [30] who noted that the failure of safety management was mostly due to the lack of safety information required for safety decision making. EBS theory proposes a new paradigm of safety management. On the one hand, it is feasible to employ EBS theory to mine useful safety information from safety data. Such safety information can be used to support making effective safety decisions, and can improve the pertinence and effectiveness of accident prevention countermeasures. On the other hand, it proposes a systematic feedback mechanism that can provide continuous safety improvements for organizations to achieve sustainable safety. It was applied to chemical accident [31] prevention, and proved to be effective.

For the above reasons, seeking a new approach to obtaining safety evidence from safety data to assist in safety decision making, and achieve sustainable safety of a system, is important for at least two reasons. First, through the above reviews, it is a research trend to use automatic information processing technologies to obtain more valuable and necessary safety information in the face of large-scale safety data. This not only helps us automatically and quickly process large-scale unstructured safety accident data, but also helps us to obtain the necessary safety information, which greatly improves the efficiency of safety data processing. Moreover, combining this with the latest safety theories allows us to uncover safety evidence more purposefully. Therefore, this study has some theoretical significance. Second, to verify the feasibility of the framework proposed in this study, we apply it to destructive coal-mine gas accidents. This will not only help us to uncover potential safety information in coal-mine gas accident data, but it will also help in reducing the occurrence of accidents, and human casualties. In addition, it provides inspiration for the mining of accident data in other fields. Therefore, this study also has some practical significance.

For this study, to automatically mine and obtain the latent safety evidence in accident data—and improve the pertinence and sustainability of safety management—a novel and practical framework to automatically extract valuable information in accident data is proposed. Owing to the severity of coal mine gas explosion accidents, a case study of accident reports of coal-mine explosions in China is carried out to illustrate the approach. The subsequent sections are organized as follows: the theoretical basis, methodology, and data sources in Section 2; the results and analysis in Section 3; and the conclusion in Section 4.

## 2. Theory and Methodology

### 2.1. Evidence-Based Safety Theory

To achieve system safety, many safety management methods have been invented. The existing safety management methods can be classified into seven types according to their connotation, and decision logic [30,32]. However, still, they often fail and lead to accidents. For example, the experience-based management method is applicable to solving similar accident problems, rather than new problems. In this case, safety decisions are made mostly from unsystematic manual experience, intuition, and simple imitations, which may make it hard to ensure the effectiveness of safety decision making. Similarly, the theory-based safety management method has poor operability and pertinence [33]. All in all, the traditional safety management research paradigm not only hinders the improvement of the system safety management level, but also fails to provide effective solutions to new safety problems in the system [34]. Therefore, the existing safety management methods are questionable [35], and new safety management approaches are urgently needed.

It is a universal and common problem in many fields that inadequate necessary information may result in decision-making failures. So, evidence-based practice (EBP) was proposed to provide a new approach and insight to address the problem of the inadequate necessary information, for better results [36]. EBP first originated in the medical field [37].

Whereafter, it was institutionalized [38] to reduce subjective physician judgment and empirical variation in the medical process, as well as to integrate the best evidence to assist in clinical decision making [37]. EBP was later applied to other fields, such as educational strategies [39,40], and evidence-based management [41]. It proved to be an effective approach. Subsequently, EBP was introduced to the field of safety management. Therefore, EBS theory was proposed. One new and significant principle EBS adds to existing safety management methods is scientific evidence. EBS creates a feedback system [22], which facilitates the sustainable operation of the entire safety management process. In other words, the core of EBS theory is to analyze and obtain safety information that produces positive safety effects, i.e., the best safety evidence, and to emphasize safety decision making based on the best safety evidence, i.e., EBS decision making. Safety evidence is defined as the findings of formal studies or investigations with any sort of scientific technique or method [22]. Safety evidence comes from safety data. Therefore, in general, safety evidence can be obtained from high-quality safety reports and research [37], as well as authoritative and comprehensive accident reports issued by relevant government departments [42].

In the era of big data, big data are regarded as the most important resource for safety management [43]. There is a wealth of value in the data that needs to be discovered. Extracting useful knowledge, actionable insights, and hidden correlations can contribute greatly to enhancing the decision-making process [44]. There is an encompassing relationship between safety big data, and safety evidence. The acquisition of the best safety evidence from the perspective of big data is shown below.

As depicted in Figure 1, the best safety evidence of coal-mine gas explosion accidents is derived from coal-mine gas explosion safety big data. After that, the best safety evidence is obtained to assist in coal-mine gas EBS decision making by utilizing data mining technologies. The decision result will also become an important basis for future analysis. It can be seen that big data can provide comprehensive best safety evidence for EBS decision making.

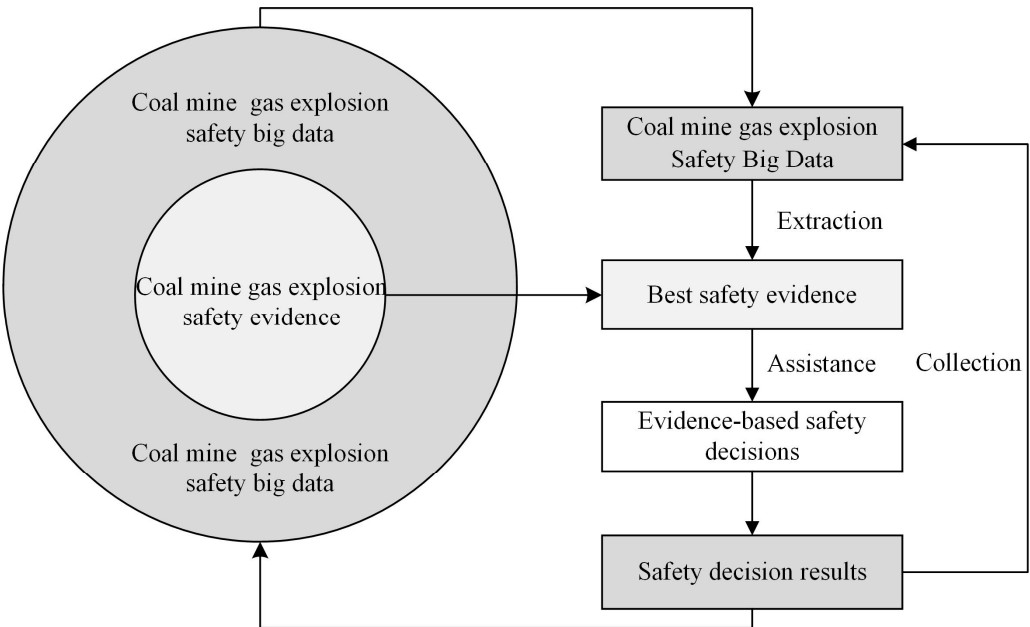

**Figure 1.** Sustainable source of best safety evidence.

Different aspects of safety evidence can be acquired by mining different levels of safety data to make different EBS decisions. In light of that, an enhanced explosion prevention and control mode optimized by best safety evidence via data mining is proposed, as demonstrated in Figure 2.

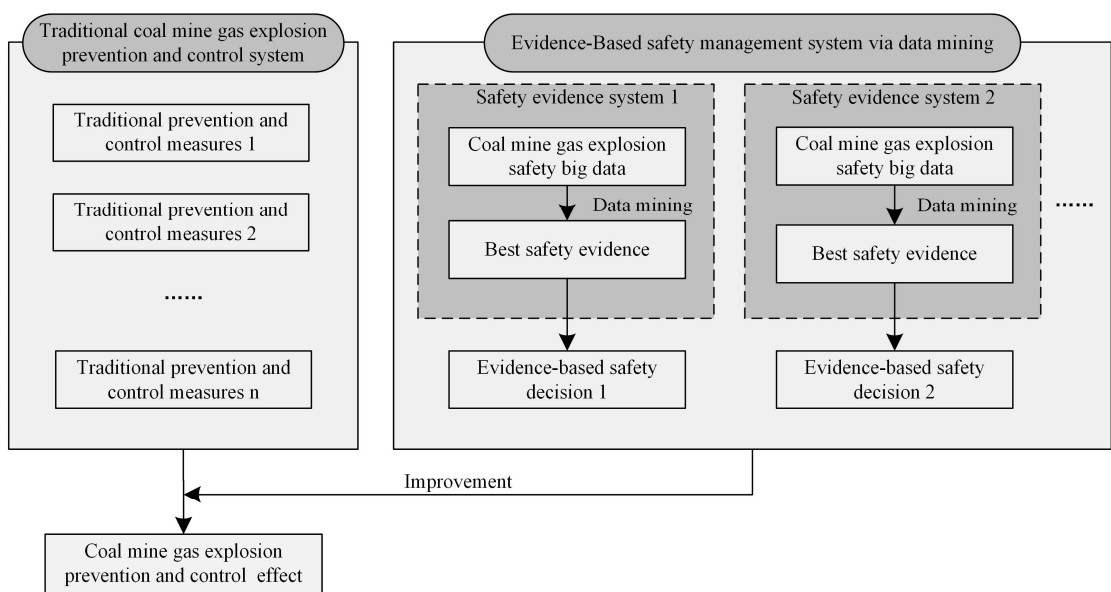

**Figure 2.** Gas explosion prevention and control mode optimized by best safety evidence via data mining.

For this study, the specific safety evidence system consists of safety evidence based on keyword extraction, safety evidence based on LDA topic mining, and safety evidence based on topic keyword position distribution. The aforesaid EBS safety decisions can, theoretically, optimize the traditional coal-mine gas explosion prevention and control mode, while improving the reliability and effectiveness of coal-mine gas safety management.

### 2.2. Data Mining Method

Data mining incorporates theories and techniques from multiple fields, such as artificial intelligence, database technologies, pattern recognition, machine learning, statistics, and data visualization, with the goal of discovering novel patterns and knowledge [45]. Currently, a substantial volume of information on the web exists in the form of unstructured and semi-structured text data [46]. As an important branch of data mining, text mining is also known as text data mining. It mainly focuses on text data, involving the conversion of unstructured and semi-structured data from large-scale text databases into digital data to extract potential knowledge. Common applications of text mining are association rule mining, text semantic mining, and text clustering analysis. The general process of text data mining includes text pre-processing, feature processing, and model construction. Since Python is an open and free platform for users and it shows great advantages in data processing, statistical analysis, model construction, and graphical presentation with its various software packages, the whole study utilizes Python and its software packages to perform the analysis of data mining.

#### 2.2.1. Text Pre-Processing

First, word segmentation processing was required. Using Python's jieba package, Chinese word segmentation was performed on sentence sequences to obtain words with smaller granularity. To avoid segmenting the professional words and to improve the accuracy of word segmentation, it was necessary to consider the professional terms of the professional fields. Therefore, the coal-mining domain and safety engineering terms from Baidu thesaurus and Sogou thesaurus were collected to construct the domain word lists. The domain word lists were continuously improved by observing the word segmentation results during the word segmentation experiments. Second, stop word processing was needed. In addition to professional words in accident reports, there were also lexical items that were meaningless for accident text analysis. To eliminate the noisy lexical items and

improve the accuracy of word segmentation, stop word lists needed to be developed. The stop word lists of Baidu, Harbin Institute of Technology, and Sichuan University were collected to construct comprehensive stop word lists, as shown in Table 1.

**Table 1.** List of domain words and stop words.

| Professional Fields | Vocabulary |
| --- | --- |
| coal-mining field | ventilators, coal dust, electric sparks . . . |
| safety engineering field | unlicensed work, safety management, safety supervision . . . |
| stop word lists | since, both, following, plus . . . |

### 2.2.2. Keyword Extraction Algorithms

The keywords can briefly summarize the document, so as to help safety managers and decision makers quickly recognize the main information hidden in the corpus of large-scale coal-mine gas explosion accidents. The TF-IDF and TextRank algorithms were used to calculate the weights. The principle of the TF-IDF algorithm [47] is shown in Equation (1).

$$TF - IDF_i = \frac{w_i}{W} \times \log\left(\frac{D}{D_w + 1}\right), \tag{1}$$

where $TF - IDF_i$ denotes the Term Frequency-Inverse Document weight of the word $i$, $w_i$ is the frequency of the word $i$ in the specified accident document, $W$ is the total number of phases in the specified accident document, $D$ is the total number of accidents in the corpus, and $D_w$ is the number of accident documents with the feature words.

TextRank is a graph-based unsupervised algorithm. The principle of the TextRank algorithm is shown in Equation (2).

$$WS(V_i) = (1 - d) + d \times \sum_{V_j \in In(V_i)} \frac{W_{ji}}{\sum_{V_k \in Out(V_j)} W_{jk}} WS(V_j), \tag{2}$$

where $WS(V_i)$ denotes the weight of the sentence $i$, $In(V_i)$ is defined as the set of words pointing to the word $V_i$, $Out(V_i)$ is the set of words pointing to the word $V_i$, $W_{ji}$ denotes the similarity of two sentences, $WS(V_j)$ represents the weight of the sentence $j$ from the last iteration, and $d$ is generally 0.85 [48].

In this paper, the TF-IDF and TextRank algorithms [21] were fused to extract the first 500 keywords from the corpus, respectively. There would be an overlap in the process of extraction. Denoted as sets $TF = \{tf_1, tf_2, \ldots\ldots, tf_{500}\}$, $TR = \{tr_1, tr_2, \ldots\ldots, tr_{500}\}$, respectively, this paper considers that if the phrase appears in both sets at the same time, the phrase is considered statistically significant in the whole text. The specific rules are shown in Equation (3).

$$tf\_tr\_w_k = \eta \times tf\_w_k + (1 - \eta) \times tr\_w_k, \tag{3}$$

where $tf\_tr\_w_k$ denotes the comprehensive weight of the word, $tf\_w_k$ denotes the weight of the keyword $k$ obtained by the TF-IDF algorithm, $tr\_w_k$ denotes the weight of the keyword $k$ obtained by TextRank, and $\eta$ takes the value of 0.5 [49].

### 2.2.3. Latent Dirichlet Allocation Model

The latent Dirichlet allocation (LDA) [50] model is a document generation model, which belongs to unsupervised machine-learning methods, and can effectively mine latent topic information. The model is a three-layer Bayesian model, including a document layer, a topic layer, and a feature word layer. In the topic model, the document set can be represented as the probability distribution of the latent topics, and the latent topics can be

represented as the probability distribution of the feature words. In this way, as is depicted in Figure 3, the document creation of LDA can be described with a two-step process [51]:

(1). For each document $i \in M$ sample a topic proportions $\theta_i$ from Dirichlet distribution $Dir(\alpha)$, and for each topic $k \in K$ sample a word proportions $\varphi_k$ from Dirichlet distribution $Dir(\beta)$.

(2). For each placeholder $j \in N$ in the document $i$

    a. Choose a topic $z_{i,j}$ randomly according to the sampled topic proportions $\theta_i$.

    b. Choose a word $w_{i,j}$ randomly according to the multinominal distribution $\varphi_k$ of the previously chosen topic $z_{i,j}$.

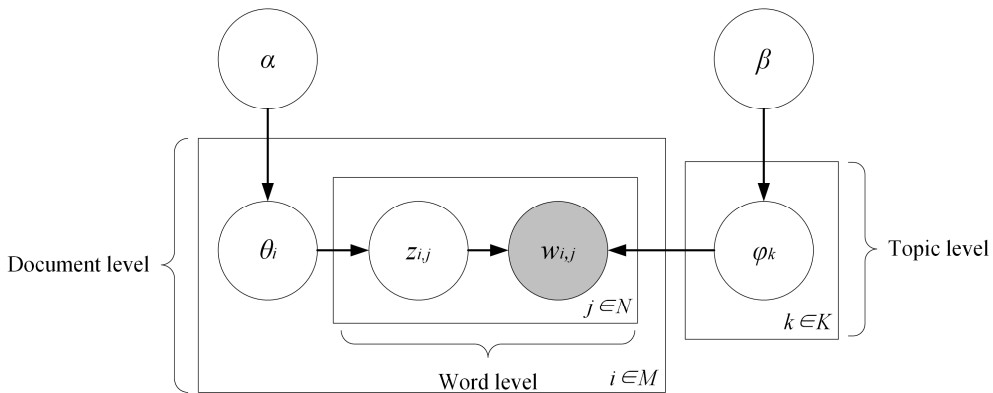

**Figure 3.** LDA structure diagram.

In the above-mentioned process, the hyperparameter $\alpha$ determines the Dirichlet prior on $\theta$ as a set of topic multinomial distributions for all documents as $M$. The hyperparameter $\beta$ determines the Dirichlet prior on $\varphi$ as a set of word multinomial distributions in all topics as $K$. Hyperparameters $\alpha$ and $\beta$ can be set to reasonable defaults [51], such as $\alpha = 1/K$ and $\beta = 1/K$ [52].

To investigate the distribution of accident causation topics, this study adopted the method of *perplexity* to determine the topics [53]. The model's capacity for generalization increases as *perplexity* decreases. The *perplexity* formula is shown in Equation (4).

$$perplexity = \exp\left\{ -\frac{\sum_{i=1}^{M} \log(P(w_i))}{\sum_{i=1}^{M} N_i} \right\}, \tag{4}$$

where $i$ denotes the document $i$, $P(w_i)$ denotes the probability of feature words in the document $i$, $M$ denotes the number of documents in the corpus, and $N_i$ denotes the total number of feature words in the document $i$.

### 2.3. Data Sources

In the process of safety production, a large amount of accident data has been accumulated. These data can be divided into structured numerical data and unstructured textual data. The unstructured safety data are often recorded in detail in textual form. In the field of safety production, accident investigation reports, as a carrier of accident text records, contain a wealth of safety information.

Therefore, in this study, a total of 202 investigation reports of coal-mine gas explosion accidents from 2000 to 2020 in China were collected as experimental data. The data were mainly obtained from the National Mine Safety Administration and other provincial bureaus of national mine safety administrations of the People's Republic of China.

*2.4. Technical Flow Chart*

In line with the aforementioned theory and methodology, Figure 4 shows the overall technical flow chart of the study, starting from the data pre-processing to the safety decision making.

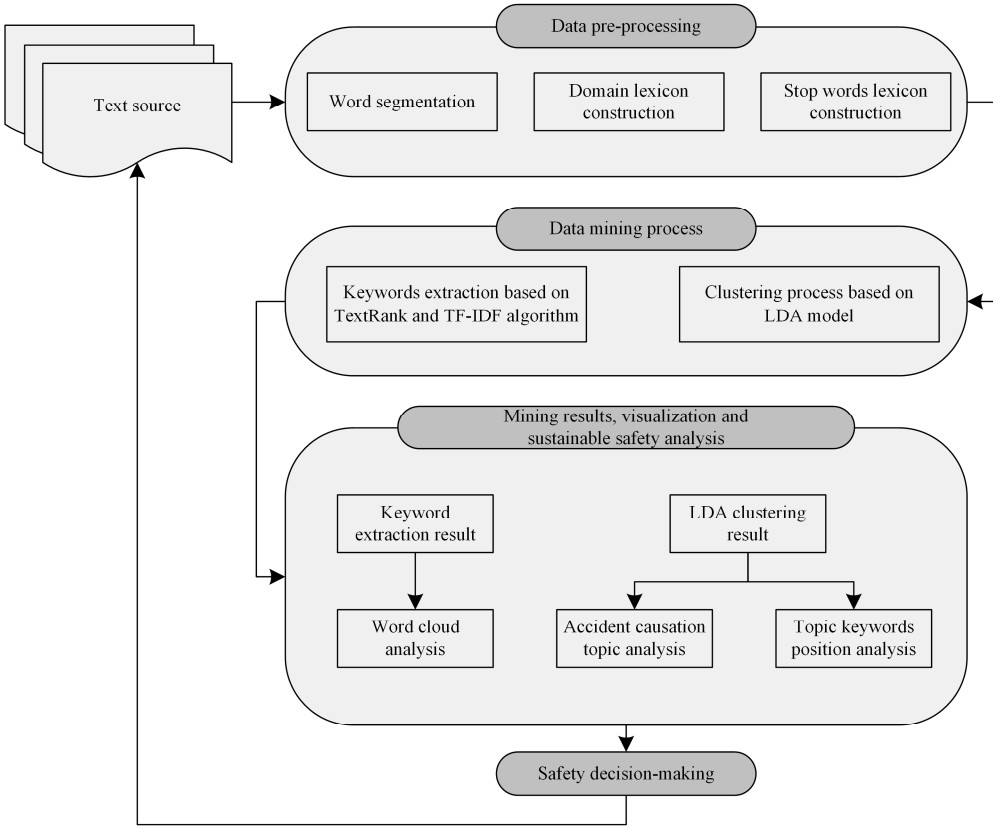

**Figure 4.** Overall flow chart of data mining based on EBS.

First, the data were collected according to the sources stated in Section 2.3. Then the collected data were pre-processed, which included word segmentation, domain lexicon construction, and stop words lexicon construction. After that, we mined the pre-processed accident data, which included extracting keywords using the TF-IDF and TextRank algorithms, and clustering the accident texts using the LDA model to explore the accident causation topic, and the specific causation factors under the topic. Furthermore, according to the above results, a visualization analysis was performed to obtain the safety evidence required for decision making. As the system safety data continues to accumulate, the results of the most recent data mining can serve as the data source for the subsequent round of data mining.

## 3. Results and Discussion

In this section, we primarily employed the methodology and data, stated in Section 2, to acquire keyword extraction and topic clustering results. At the same time, the results were visualized, and the analysis was carried out.

### 3.1. Keyword Extraction Result and Analysis

According to the TF-IDF and TextRank algorithms, keywords of accident texts from 2000 to 2020 were extracted. The main concerns of the accidents could be initially derived using Equations (1)–(3). As shown in Table 2, the top 10 corresponding keywords are "management disorder", "safety supervision", "ventilation fan", "violation of regulations",

"regulatory authority", "illegal organization", "ventilation", "employees", "explosion", and "safety awareness".

**Table 2.** Distribution of keywords.

| Keywords | Combined Weights | Keywords | Combined Weights |
|---|---|---|---|
| management disorder | 0.23662 | management | 0.05841 |
| safety supervision | 0.11951 | working face | 0.06336 |
| ventilation fan | 0.08934 | discharge | 0.06121 |
| violation of regulations | 0.08777 | electric sparks | 0.05976 |
| regulatory authority | 0.08292 | detonation | 0.05059 |
| illegal organization | 0.07797 | sparks | 0.05384 |
| ventilation | 0.08270 | illegal operation | 0.04818 |
| employees | 0.06911 | production | 0.04900 |
| gas explosion | 0.06653 | operation | 0.04810 |
| safety awareness | 0.06428 | localized aggregation | 0.04343 |

The visualization of accident text keywords, using word cloud mapping technology, [54] can help safety managers to capture the main accident causation factors in a large number of accident texts more quickly and intuitively. The word cloud package in Python was used to visualize the keywords and produce word cloud evolution graphs every 5 years, as shown in Figure 4. The font size in the graph is positively correlated with the comprehensive weight of the keywords in the corpus.

As can be seen from Figure 5, representative keywords do not significantly change over time, and accident causation factors share commonality and universality. They are mostly clustered in the chaotic management, the inadequate supervision by the supervisory department, and the failure of the ventilation system.

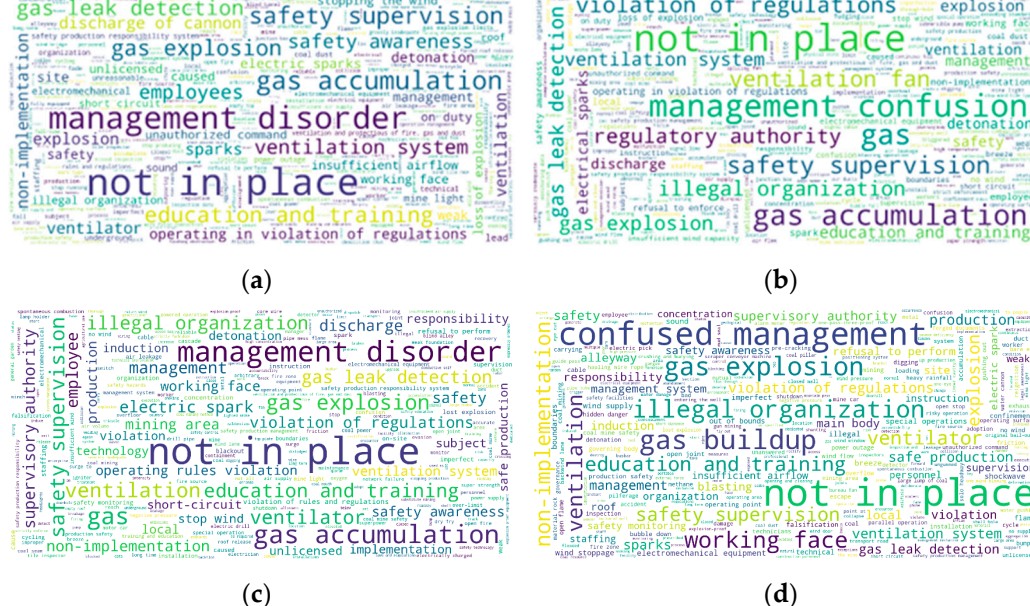

**Figure 5.** Word cloud evolution from 2000 to 2020: (**a**) 2000–2005; (**b**) 2006–2010; (**c**) 2011–2015; (**d**) 20016–2020.

Therefore, the best safety evidence obtained from keywords and word cloud evolution graphs shows that failures of management and ventilation are important causes of accidents.

In the process of operation, enterprise managers should clarify the common and universal problems of the coal-mine gas safety management, especially for the management of ventilator-related equipment, and the management of staff violations. Employees should never work against the rules. Safety managers should strengthen safety education for employees, enhance their safety knowledge and safety awareness, and take corresponding incentive or punishment mechanisms to reduce staff violations. Relevant supervisory departments should pay attention to the safety supervision of coal-mine enterprises' gas conditions, and implement relevant regulations.

### 3.2. LDA Clustering Results and Analysis

3.2.1. Accident Causation Topic Analysis

The LDA model was constructed using Python, and accident topics were mined at a deeper level. The *perplexity* was calculated by Equation (4). It was found that when the number of total topics $K = 4$, the *perplexity* was the lowest, which meant the accident text topic result had a good differentiation. The number of accident causation topic keywords was 15, as shown in Table 3. Topics were visualized with pyLDAvis toolkit, as shown in Figure 6.

As can be seen from Figure 6, the circle on the left side represents the results of topic clustering with four tags. The size of the circle is positively related to the number of documents that are subordinate to the topic. The bars on the right represent the 30 terms that are most relevant to the topic. We can see that Topic 1, and Topic 2 have the highest number. Topic 1 has a strong correlation with Topic 3. Combined with Table 3, the specific analysis of accident causation topic analysis and safety decision making is as follows.

(1) Topic 1: Organizational supervision and personnel management. As can be seen from Table 3, the main problems are mainly composed of the insufficient supervision by the supervisory department, the failure to carry out the main responsibility, the imperfection in rules and regulations, insufficient staffing, and the absence of on-site inspections. Keywords such as "not in place", "unreasonable", and "responsibility" indicate that the safety awareness of enterprise managers is not sufficient. Although the relevant rules and regulations have been formulated, and safety education and training have been conducted for employees, the relevant procedures fail to go deep and thorough. Quantitatively speaking, inadequate safety management and staffing are important causes of accidents.

(2) Topic 2: On-site production environment management. As seen from Table 3, this topic is mainly manifested by illegal discharges, circulating wind leading to gas accumulation, cable short-circuits, and coal dust. Keywords such as "violation", "operation", "ventilation", "cable", "coal-dust ", and "electric sparks" demonstrate that ventilation problems tend to cause gas accumulation, while cables tend to cause electric sparks to ignite the accumulated gas. The high numbers related to this topic also show that this is the most direct cause of coal-mine gas explosions. Therefore, attention should be focused on the management of equipment and substances such as fans, coal dust, and cables on the site to avoid gas accumulation and ignition.

(3) Topic 3: On-site production equipment management. This topic mainly consists of air leakage from fans, the detonation of electromechanical equipment, and the malfunction of gas monitoring systems. This topic has a strong correlation with Topic 1. Cutting off the elements that are related to Topic 1 and Topic 3 can reduce the occurrence possibility of accidents. For example, more attention should be paid to improving and standardizing the operation, as well as management procedures of electromechanical equipment, ventilation equipment, and monitoring equipment. The intensity and frequency of on-site inspections should be strengthened. At the same time, safety and technical education for front-line employees should be more frequent.

(4) Topic 4: Production process and hazardous substances. This topic mainly relates to gas accumulation in mining areas, working faces, roadways, electric sparks generated by illegal operations, and mine light malfunction. Keywords such as "gas", "concentration", and "explosion limits" indicate that Topic 4 focuses more on production processes

and hazardous materials. Therefore, during the production process, it is necessary to ensure the normal operation of the ventilation system to avoid the accumulation and high concentration of gas.

**Table 3.** Topic extraction results.

| Topic Number | Topic Keyword | Summary of Causation Topics |
|---|---|---|
| Topic#1 | safety management, employee, violation, supervisory department, technical skills, on-site management, on-site inspection, production, staff management, supervision, responsibility, regulations, management system, unreasonable | Organizational supervision and staff management topic: (1) supervisory department supervisions are not in place; (2) the responsibility is not taken; (3) rules and regulations are not comprehensive; (4) the safety education for staff is not enough; (5) the on-site inspection is not in place. |
| Topic#2 | violation of regulations, illegal work, discharge, localized aggregation, ventilation, sparks, electric sparks, work face, circulation, staff, short-circuit, flames, cable, explosion limits, coal dust | On-site production environment management topic: (1) violation of regulations such as unauthorized discharge; (2) circulation wind leading to gas accumulation; (3) sparks from short-circuit cables; (4) coal dust involved in explosions. |
| Topic#3 | technical skills, mechanical, fan, mine light, electrical equipment, wind leakage, on-site management, on-site inspection, monitoring, roadway, ventilation, sparks protection, monitoring system, unreasonable, fault | On-site production equipment management topic: (1) wind leakage and ventilation fault; (2) electrical and mechanical equipment fault; (3) gas monitoring system failure. |
| Topic#4 | work face, concentration, sparks, coal mine gob, electric sparks, roadway, mine lights, explosion limits, workers, gas, wind leakage, violation, localized aggregation, short circuit, status | Production process and hazardous material topic: (1) gas accumulation in the mining area, working face, and roadway; (2) gas reaching the explosion limit; (3) illegal operation generating electric sparks; (4) mine lamp malfunction. |

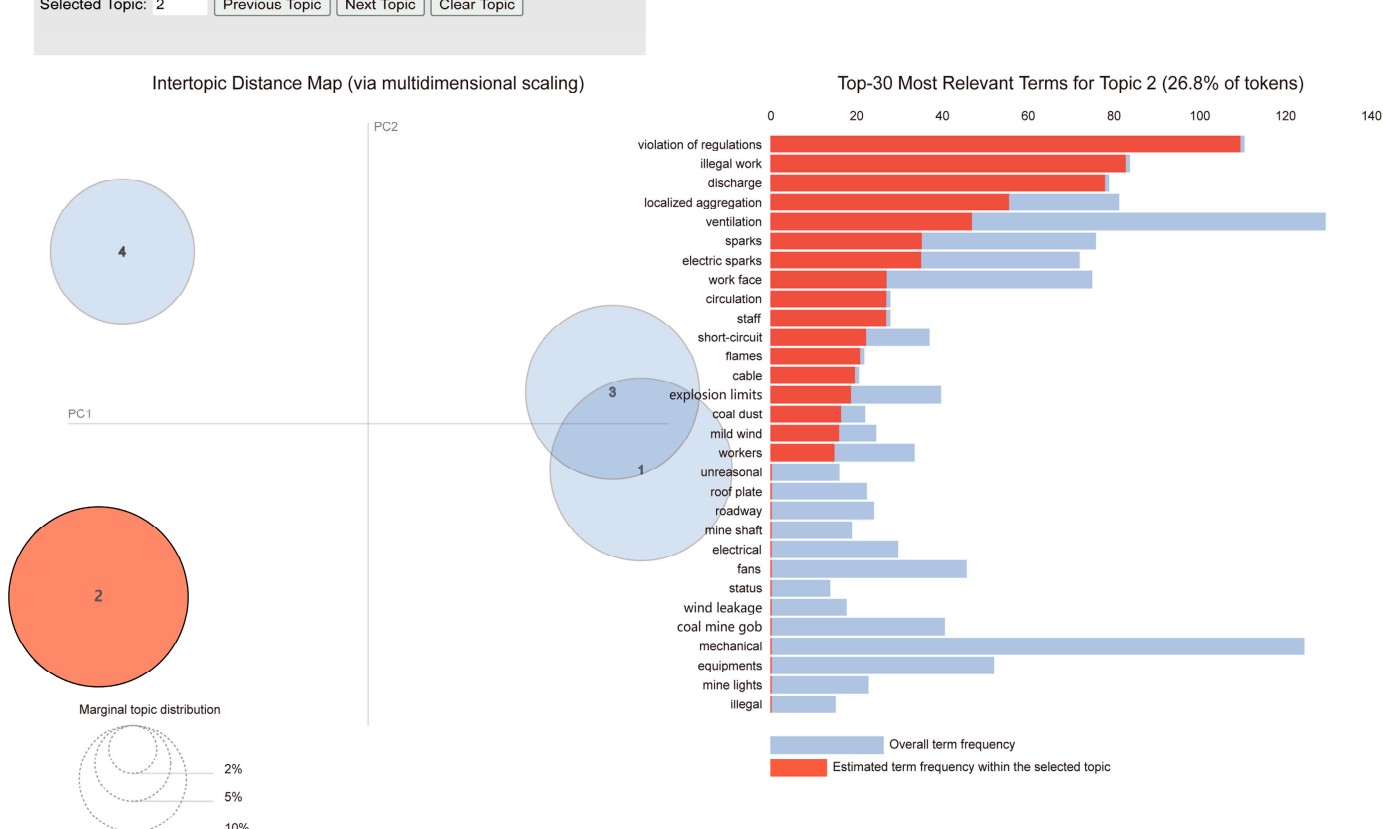

**Figure 6.** Visualization of topic distribution.

The aforementioned topic mining analysis reveals four coal-mine gas accident topics related to organizational supervision and personnel management, on-site production environment management, on-site production equipment management, and production process and hazardous substances. Additionally, organizational management topics and on-site

management topics occupy a larger proportion, while the organizational management topic has a strong correlation with the production equipment topic. As a result, strengthening the improvement of management systems should be the primary emphasis of accident prevention, along with the regular inspection of production equipment.

### 3.2.2. Topic Keywords Position Analysis

To further understand the location distribution of topic keywords in the corpus, the corpus accident texts were connected from the beginning to the end. The results were numbered to draw a topic vocabulary distribution map, which represented the vocabulary distribution location more intuitively and clearly.

As shown in Figure 7, the horizontal coordinate represents keyword positions, while the vertical coordinate represents different topic terms. Since the corpus is organized chronologically, topic keywords position distribution can also be seen as a temporal distribution. The shorter vertical lines indicate the position of each term. The density of the short vertical line can reflect the frequency of the term.

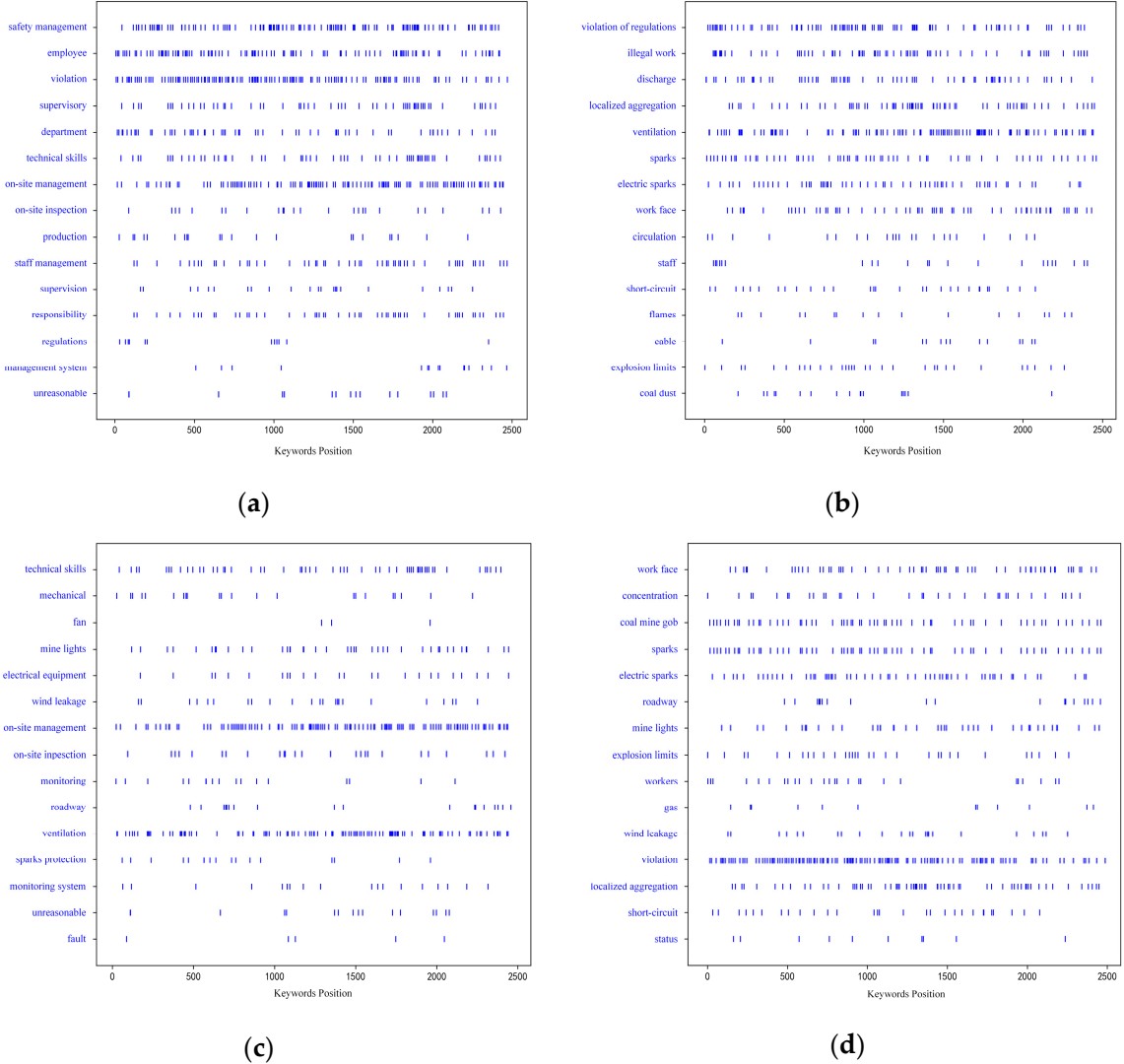

**Figure 7.** Visualization of topic keywords positions. (**a**) Topic 1: Organizational supervision and personnel management; (**b**) Topic 2: On-site production environment management; (**c**) Topic 3: On-site production equipment management; (**d**) Topic 4: Production process and hazardous substances.

According to the topic terms obtained from the accident topic mining analysis, respectively, it is found that terms related to safety management are more concentrated in

Figure 7a. Terms related to on-site risk management are more concentrated in Figure 7b. In Figure 7c, terms related to management and ventilation are more concentrated. This suggests that failures in enterprise safety management, supervisory department supervision, and on-site production environment management are frequently discussed, so there are problems in the implementation of relevant prevention and control.

Therefore, for coal-mine gas explosion accident prevention, enterprises should focus on the implementation of the main responsibility—the improvement of the rules and regulations, and the implementation of supervision and supervision departments. When carrying out on-site risk management, safety managers should focus on the staff's illegal operations. Priority should be given to strengthening the on-site management of ventilation fans, electromechanical equipment, and coal dust, to avoid the accumulation of gas caused by the abnormal operation of fans, as well as electric sparks produced by short circuits.

### 3.3. Sustainable Safety Analysis

Through the above analysis, valuable safety information can be initially obtained, and relevant safety decisions can be made. However, for the sustainable safety management of complex systems, a better mechanism is required to guarantee that safety information can be used sustainably. For this reason, based on EBS theory, a feedback mechanism is introduced. As is shown in Figure 1, all of the above safety evidence, that is discovered through data mining, will theoretically be incorporated into the new safety big data. As the safety big data expand, they can be used for future data mining work with other data mining techniques such as Bayesian network, decision tree (DT), support vector machine (SVM), and k-nearest neighbor (KNN). In this way, new safety information can be acquired to achieve the sustainable safety of a system.

## 4. Conclusions

Data mining in coal-mine gas explosion accidents enables us to gain useful safety information from past accidents, and better define future safety prevention and control priorities.

The primary contribution and novelty of this study are outlined as follows. A novel data mining framework to automatically extract valuable information in accident data, based on EBS, was proposed. It aimed to improve the pertinence and effectiveness of safety management and achieve the sustainable safety of a system, which could help safety decision makers to achieve evidence-based safety management in engineering management. TF-IDF and TextRank algorithms were combined to extract keywords from the accident corpus. By analyzing the evolution of text keyword word cloud of coal-mine gas explosion accidents through the time dimension, the analysis revealed that accident causation factors shared commonality, and universality. The LDA model was applied to mine accident text topics. By mining accident causation topics from large-scale accident corpus, and obtaining the causation factors under different accident topics, it helped decision makers take preventive and corrective measures in a targeted manner. It can be concluded that data mining, based on EBS, can answer the imminent need to turn safety data into useful safety evidence, realize data-driven EBS management, and improve the reliability and sustainability of EBS decision making, thus achieving the process safety and essential safety of a system.

The limitations of the study, and possible improvements for a future study, can be considered. The above study demonstrates that data mining in coal-mine explosion accidents, based on EBS, can delve further into accident causation topics and specific causation factors. Nevertheless, owing to the relatively singular data source and uniform text data structure, the model is limited in its capacity to adapt to the text data of various structures. Given different corpus and data mining tasks, the extraction of valuable information varies a lot [55].In future, it can be improved and applied to more text data with different sources, such as the accident query record system of a specific coal mining company, and the Internet. Moreover, further research can be conducted in other engineering fields, such as construction engineering, and chemical engineering.

**Author Contributions:** Conceptualization, J.H. and R.H.; methodology, J.H.; formal analysis, F.X.; writing—original draft preparation, J.H. and R.H.; writing—review and editing, F.X.; funding acquisition, R.H. All authors have read and agreed to the published version of the manuscript.

**Funding:** This research was funded by the National Key R&D Program of China (2018YFC0808406) and the Fundamental Research Funds for the Central Universities of Central South University (2022ZZTS0487).

**Institutional Review Board Statement:** Not applicable.

**Informed Consent Statement:** Not applicable.

**Data Availability Statement:** Not applicable.

**Conflicts of Interest:** The authors declare no conflict of interest.

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
