# Peer review of "Data Mining in Coal-Mine Gas Explosion Accidents Based on Evidence-Based Safety: A Case Study in China"

_sustainability, doi:10.3390/su142416346_

Round 1

Reviewer 1 Report

1-replication of words: line (6) Correspondence: Correspondence

2- replication problem in the first two sentences (line 7 and 8)

3-The problematic of this work is not clear in the abstract

4-theoretical background is insufficient

5- The space between the text and the figures/tables must be respected 

6-poor quality of figures 6 and 7.

7- the results are not well represented and not clearly interpreted.

8- in the conclusion, use paragraphs instead of listings 

9-plagiarism: Your Document needs Optional Improvement. 

Reviewer 2 Report

The present manuscript describes a case study of the data mining in coal mine gas explosion accidents. A manuscript has a practical application and also provides important theoretical for the next studies. The results show that the accident data mining based on evidence-based safety can effectively and purposefully mine the best safety evidence, and guide the safety decision-making to optimize the present safety management model and achieve sustainable safety.

The paper can be accepted for publication after providing the corrections mentioned below.

Comment 1. The Introduction section is really short. In the Introduction section, an enhanced literature review is required. For this study, the authors have used only 22 references. It seems insufficient for such type of research. Moreover, most references are from the China.

Comment 2. You should provide more deeper analysis of importance to conduct your study based on the provided information about gas explosion. In this case I recommend using below mwntioned reference for analysis”.

Zavialova, O., Kostenko, V., Liashok, N., Grygorian, M., Kostenko, T., & Pokaliuk, V. (2021). Theoretical basis for the formation of damaging factors during the coal aerosol explosion. Mining of Mineral Deposits, 15(4), 130-138. https://doi.org/10.33271/mining15.04.130

Important issue: “Over the past four decades, about three dozen explosionsof air-methane mixtures or hybrid mixtures of coal dust with methane and air have occurred in Ukrainian coal mines (Table1)”.

Comment 2. It will be great if the authors show some description in context – Why it is important to conduct this study? Can the expected result be used or implemented within other simulated conditions? If yes, then how? What limitations?

Comment 3. The aim and the tasks must be highlighted at the end of the Introduction section.

Comment 4. Authors indicate: “Wang [22] first proposed the theory of Evidence-Based Safety (EBS), pointing out that the failure of safety management is mostly due to the lack of safety information required for safety decision-making”. The question is – why authors select EBS as a tool in their research. Explanation is required.

Comment 5. In the subsection Technical Flow Chart please add more details and discussion of the flow chart of data mining based on EBS.

Comment 6. Figure 7. Sharper figures are required.

Comment 7. Please provide a short description of further research.

Comment 8. The novelty of the paper must be highlighted in the conclusions section.

Comment 9. Please consider the suggested research in your paper when enhancing the literature review* (enhanced review of non-China authors only are more than welcome):

Kobylianskyi, B., & Mуkhalchenko, H. (2020). Improvement of safety management system at the mining enterprises of Ukraine. Mining of Mineral Deposits, 14(2), 34-42. https://doi.org/10.33271/mining14.02.034

Important issue: “In the mentioned paper is proposed to improve the safety system by introducing a “smart-protection” system, which is triggered at the stage of hazards identification, increasing the decision-making adequacy”

Comment 10. The content of the manuscript is similar to that of a case study. The knowledge contained here may be useful for engineers, students, and scientists, searching for any knowledge related to mining engineering, which is the most important value of the manuscript. In general, I must admit that a very good study was performed, and I will recommend your paper for publication after careful revision.

Reviewer 3 Report

 Aceite após pequenas correções 

Round 2

Reviewer 1 Report

all remarks are satisfied 

Reviewer 2 Report

Dear authors,

I am more than satisfied with the corrections provided by you.

This study is an important contribution to sustainable mining.

Congratulations to the authors.